# Exploring the Cyclic Behaviour of URM Walls with and without Damp-Proof Course (DPC) Membranes through Discrete Element Method

**Bora Pulatsu** [1,*] **, Rhea Wilson** [1] **, Jose V. Lemos** [2] **and Nebojša Mojsilović** [3]

1    Department of Civil and Environmental Engineering, Carleton University, Ottawa, ON K1S 5B6, Canada; rheaswilson@cmail.carleton.ca
2    National Laboratory for Civil Engineering (LNEC), Av. do Brasil 101, 1700-066 Lisbon, Portugal; vlemos@lnec.pt
3    Institute of Structural Engineering, Department of Civil, Environmental and Geomatic Engineering, ETH Zurich, 8093 Zurich, Switzerland; mojsilovic@ibk.baug.ethz.ch
*    Correspondence: bora.pulatsu@carleton.ca

**Abstract:** Unreinforced masonry (URM) walls are common load-bearing structural elements in most existing buildings, consisting of masonry units (bricks) and mortar joints. They indicate a highly nonlinear and complex behaviour when subjected to combined compression–shear loading influenced by different factors, such as pre-compression load and boundary conditions, among many others, which makes predicting their structural response challenging. To this end, the present study offers a discontinuum-based modelling strategy based on the discrete element method (DEM) to investigate the in-plane cyclic response of URM panels under different vertical pressures with and without a damp-proof course (DPC) membrane. The adopted modelling strategy represents URM walls as a group of discrete rigid block systems interacting along their boundaries through the contact points. A novel contact constitutive model addressing the elasto-softening stress–displacement behaviour of unit–mortar interfaces and the associated stiffness degradation in tension–compression regimes is adopted within the implemented discontinuum-based modelling framework. The proposed modelling strategy is validated by comparing a recent experimental campaign where the essential data regarding geometrical features, material properties and loading histories are obtained. The results show that while the proposed computational modelling strategy can accurately capture the hysteric response of URM walls without a DPC membrane, it may underestimate the load-carrying capacity of URM walls with a DPC membrane.

**Keywords:** unreinforced masonry; computational modelling; discrete element method; contact mechanics

## 1. Introduction

Unreinforced masonry (URM) walls constitute the main structural components in most existing buildings and heritage structures. While the construction morphology and material used in masonry walls vary considerably, URM walls typically comprise stone or brickwork assemblages where cement and/or lime-based mortar are used as binding materials. The strong brick–weak mortar joint combination is commonly noted in old masonry buildings, which is associated with low bond strength properties at the unit–mortar interfaces unless repointed via relatively high-strength mortar.

Past and recent post-earthquake investigations have revealed the diverse failure mechanisms of URM buildings, where cracks are predominantly localized along the unit–mortar interfaces due to weak tensile strength and shear resistance [1–5]. The out-of-plane (OOP) failure mode is consistently noted as the most common collapse mechanism, which is related to the weak wall–diaphragm connections and poor or no interlocking corner bond pattern between orthogonal walls, among many other factors [6,7]. On the other hand,

in-plane (IP) failure modes can be activated once the OOP mechanism is restrained under lateral seismic actions. Sliding, diagonal tension cracking and rocking/flexural failures are the typical IP collapse mechanisms of URM walls that may develop solely or interact with each other (i.e., mechanism changes during lateral deformation or cyclic loading) [8]. It is worth noting that wall aspect ratio, mechanical properties of masonry constituents, imposed pre-compression load, bond pattern, and size and location of the openings affect the IP load-carrying capacity and associated collapse mode of URM walls [9–12]. Given the considerable material uncertainty in old masonry buildings, in addition to the factors mentioned earlier, it is challenging to accurately predict the seismic response of URM walls. Furthermore, choosing cost-effective and efficient retrofitting solutions requires a clear understanding of their seismic response. In this context, validated advanced computational models play an important role in exploring the mechanics of URM walls and are used in the decision-making process.

Briefly, computational modelling strategies used for the structural/seismic assessment of URM buildings can be grouped into continuum and discontinuum-based approaches [13]. Traditional continuum-based models, also referred to as macro-modelling, represent masonry as a homogeneous medium by utilizing averaged constitutive relationships to define nonlocal mechanical properties and are commonly employed in the structural assessment of masonry buildings [10,14–20]. Although macro-modelling offers a computationally efficient strategy for analyzing the overall behaviour of the masonry structure, it may not accurately capture the damage localization and crack propagation influenced by the morphological features of the masonry. Conversely, discontinuum-based analysis, like the discrete element method (DEM), explicitly simulates the cohesive bond behaviour and fracture in masonry units by representing the masonry texture as a system of discrete blocks [21–23]. Each masonry unit can be considered to be a distinct single block (either rigid or deformable) or made up of multiple blocks using triangle (2D), tetrahedral (3D) or random polyhedral blocks [24,25]. Over the last few decades, different modelling techniques have been proposed within the DEM framework to analyze URM walls under quasi-static and cyclic loading. Malomo et al. (2019) proposed a DEM-based methodology using distinct elastoplastic blocks interacting with each other along their boundaries to simulate the response of URM assemblies under quasi-static cyclic loads [26]. The macro-stiffness and strength degradation of the analyzed masonry walls are captured as the contact points between blocks that fail in tension and shear. More recently, Damiani et al. (2023) investigated the in-plane cyclic performance of masonry walls with and without steel reinforcement [27]. A similar modelling strategy was followed considering a plastic material model using the Mohr–Coulomb criterion for deformable blocks, but in contrast to the previous study, fracture energy-based elasto-softening contact constitutive models were employed. The authors noted a reasonable match in both studies when comparing their results to the experimental findings.

In this research, different from earlier studies, a more sophisticated contact constitutive model embedded in the DEM framework was used to analyze the IP cyclic behaviour of URM walls, adopting various pre-compression vertical pressures. Most DEM-based studies in the literature implement standard brittle contact constitutive models in computational models without addressing the stiffness degradation in compression and tension regimes at the contact points. Furthermore, the coupling mechanisms between compression and shear are either ignored or no compression failure is defined. Therefore, this study offers the first application of a recently developed coupled elasto-softening contact model (see [28]) in analyzing URM walls under cyclic loading. The proposed modelling strategy is validated and used to simulate the experimental testing campaign presented by Mojsilović et al. (2010), which tested URM wallettes under cyclic loading, both with and without damp-proof course (DPC) membranes [29].

## 2. Computational Procedure of Discontinuum-Based Rigid Block Analysis

In this study, URM walls are analyzed following the simplified micro-modelling approach based on the discrete element method (DEM) developed by Cundall [30]. The discontinuous nature of URM walls is represented within the DEM framework, where each masonry unit is denoted by two rigid blocks with a potential crack surface, whereas the bond (or unit–mortar interface) is explicitly considered using zero-thickness interfaces. Note that the adopted modelling strategy may also be denoted as discrete rigid block analysis (D-RBA), and various applications of D-RBA can be found in the literature (e.g., [31]).

The computational procedure of the DEM utilizes a time-marching scheme to solve equations of motion, adopting the central difference method. The implemented explicit solver provides the new translational ($\dot{u}_i^{t+}$) and rotational velocities ($\omega_i^{t+}$) for rigid blocks (each has 6 degrees of freedom—3 translations and 3 rotations), given in Equation (1).

$$
\begin{aligned}
\dot{u}_i^{t+} &= \dot{u}_i^{t-} + \frac{\Delta t}{m}\left(\Sigma F_i^t - \lambda \left|\Sigma F_i^t\right| sgn\left(\dot{u}^{t-}\right)\right) \\
\omega_i^{t+} &= \omega_i^{t-} + \frac{\Delta t}{I}\left(\Sigma M_i^t - \lambda \left|\Sigma M_i^t\right| sgn\left(\omega^{t-}\right)\right)
\end{aligned}
\tag{1}
$$

where $m$, $I$ and $\Delta t$ are the block mass, moment of inertia and time step. The unbalanced force (or total force, $\Sigma F$) is calculated for each rigid block considering the gravitational load, external loads and subcontact forces. Similarly, the net/total moment ($\Sigma M$) is the sum of moments produced by contact and applied forces. Note that the new translational and angular velocities are predicted at the mid-time intervals (i.e., $\Delta t$, $t^+ = t + \Delta t$, $t^- = t - \Delta t$), and the quasi-static solutions are obtained via Cundall's local damping formulation [32]. After computing new velocities, they are utilized to update block positions and relative block displacements. Accordingly, contact forces among adjacent blocks are computed, which are a function of relative normal and shear contact displacements and used in the equations of motions in the next time step (Equation (1)).

In DEM, the mechanical interaction among the rigid blocks is predicted by calculating the action/reaction forces at the point contacts (also called sub-contacts—see Figure 1a) using three orthogonal springs (one in the normal and the other two in the shear directions). The sub-contacts are defined along the contact plane between the adjacent blocks, and overlapping at the contact points is allowed and controlled by the contact stiffness ($k_n$: normal contact stiffness, $k_s$: shear contact stiffness). A typical point contact configuration and the associated contact plane are shown in Figure 1a for a two-block system in which the bottom block is fixed and the top block is subjected to rotation about the z-axis. As mentioned earlier, sub-contact stresses in the normal ($\sigma$) and shear ($\tau$) directions are computed based on the relative displacement in the normal and shear directions ($u_n^c$, $u_s^c$), respectively, illustrated in Figure 1b.

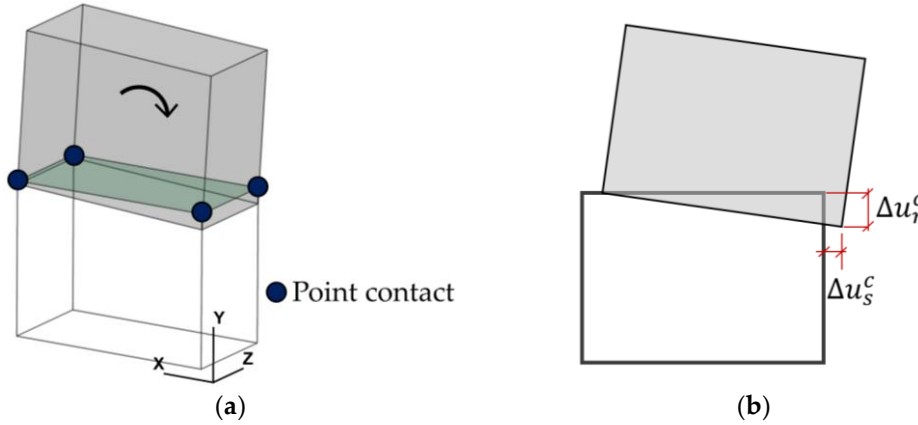

(a)  (b)

**Figure 1.** (**a**) Illustration of point contact and the contact plane defined between the two blocks and (**b**) relative point contact displacement in the normal and shear directions.

Most DEM-based computational models in the literature adopt relatively simple contact constitutive laws to capture the bond behaviour. A standard contact model (e.g., brittle failure upon exceeding the bond strength in shear and tension together with no compression failure) only requires bond tensile strength ($f_t$) for the normal direction and cohesion ($f_t$) and friction angle ($\phi$) for the shear direction in addition to contact elastic (initial) stiffnesses. In other words, only three input parameters are needed to create yield functions for tension and shear. Although it provides practicality to perform nonlinear structural analysis of URM walls and buildings (i.e., using a limited number of input parameters), the accuracy of the computational models should be carefully interpreted. As discussed recently, the standard (brittle) contact models cause abrupt changes in the macro-behaviour of the URM walls due to sudden changes in the stress state at the contact points and may yield unnecessary pre-mature failures [28]. Moreover, heavy calibration procedures may be necessary to obtain accurate predictions, especially in the absence of any similar input data set employed in relevant studies. Alternatively, more sophisticated contact models can be used to better capture the local mechanics of the unit–mortar interfaces (or bonds) by considering the post-peak response of the cohesive bond in both the tension and shear directions. Throughout this research, a recently proposed contact model is utilized, where the bi-linear elasto-softening material model is adopted in tension, compression and shear regimes. Stiffness degradation is considered at the contact point only in the normal direction. Upon unloading or reloading conditions, secant stiffness is considered, which is depicted in Figure 2. A stiffness recovery algorithm is implemented upon tensile failure to capture the crack-closure and -opening phenomena under reversal loading cycles.

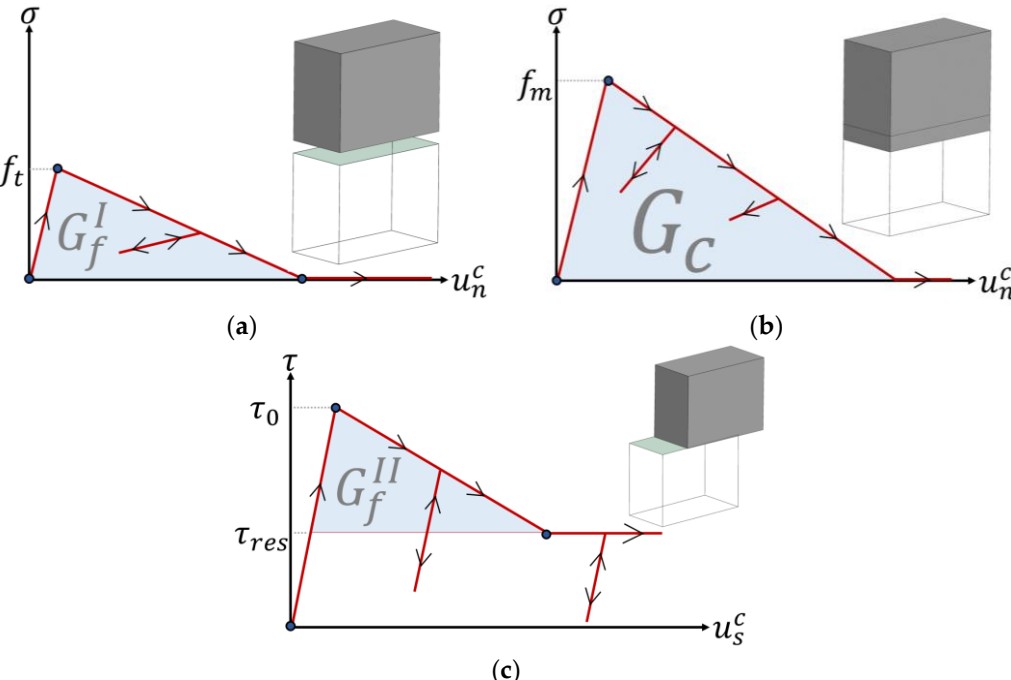

**Figure 2.** Illustration of point contact and the contact plane defined between the two blocks in (**a**) tension, (**b**) compression and (**c**) shear.

It is worth noting that within the adopted discrete rigid block modelling strategy, the crushing or compression failure of masonry composite and bond tensile and sliding shear failures are all represented at the contact point. The defined failure domain is shown in Figure 3. The shear strength is governed by the Mohr–Coulomb failure criterion ($\tau - [c - tan\theta(\sigma)] \leq 0$, $\sigma < 0 \rightarrow$ compression), whereas tension ($\sigma - f_t \leq 0$) and compression cut-off ($-\sigma - f_m \leq 0$) are used in the normal direction (see Figure 3). A two-way coupling mechanism is considered between tension and shear upon failure in

either of them, while one-way coupling is implemented between shear and compression, meaning that compression failure affects the shear capacity; however, shear failure does not influence the compression strength. The readers are referred to the reference study for further details regarding the mathematical and computational procedure of the employed contact constitutive model [28].

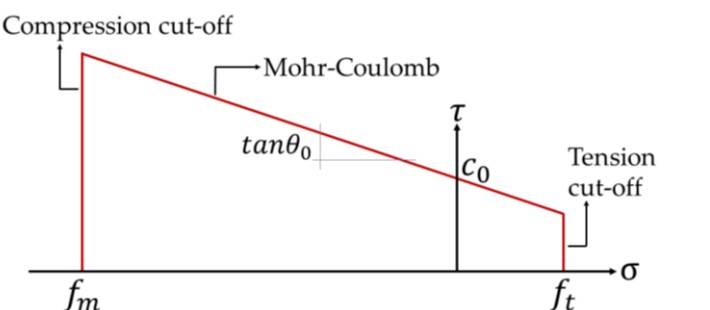

**Figure 3.** Defined failure envelope at the contact points.

The computed contact stresses are then multiplied by the associated sub-contact area and utilized in the equations of motion. The numerical stability of the explicit solver is ensured by using satisfactorily small time steps, less than a critical value $\Delta t_{cr} < 0.2\sqrt{m_{min}/k_{n,max}}$, controlled by the minimum block mass ($m_{min}$) and the maximum contact stiffness ($k_{n,max}$) defined in the discontinuous system.

A commercial three-dimensional discrete element code, 3DEC, developed by Itasca, is used to perform quasi-static analysis of URM walls under cyclic loading [33]. The explained contact model is implemented through the user-defined constitutive model option that is compiled as a DLL (dynamic link library) into 3DEC.

## 3. Validation Studies

The adopted computational modelling strategy and the coupled elasto-softening contact model are validated using the experimental testing campaign presented by Mojsilović et al. (2010) [29]. In the reference study, the cyclic behaviour of URM walls under various vertical pressures was tested with/without a damp-proof course (DPC). A DPC is commonly placed in the bed joints of URM walls (either above or below the mortar joints), providing a moisture barrier and accommodating joint sliding by allowing for differential movements.

The reference testing campaign included three series of URM walls: one control set without a DPC (Series C), another with the DPC layer between the first two masonry courses (Series A), and a third with the DPC layer placed between the masonry wall and the concrete base (Series B). For each series, three different levels of pre-compression were considered. A summary of the conducted testing campaign is presented in Table 1. For Series A and B, the tests were repeated three times at each pre-compression level. Note that Series B and C will validate the proposed modelling strategy for the current investigation.

**Table 1.** Experimental testing summary.

| Series | Applied Pre-Compression Stress (MPa) | | |
| :---: | :---: | :---: | :---: |
| | 0.7 | 1.4 | 2.8 |
| A | A3 | A1 | A2 |
| B | B3 | B1 | B2 |
| C | C3 | C1 | C2 |

The URM wallettes were constructed on reinforced concrete beams using extruded clay bricks ($230 \times 110 \times 76$ mm$^3$), 1:1:6 cement–lime–sand mortar, and embossed polythene

membranes. In each damp-proof course, the embossed polythene membrane was applied directly on the masonry or concrete base, with the mortar joint then laid on top of the membrane. Each test wallette was five units long and 14 courses tall, with a thickness of 110 mm. Note that the exact dimensions of each specimen varied slightly. Material characterization tests were also conducted to gather information about the specimens, including the compression strength of the masonry components and composite (tested on masonry triplets) and the flexural, tensile and shear bond strength of the masonry. Further details on the experimental test setup can be found in Mojsilović et al. (2009) [34]. A quasi-static cyclic loading protocol was followed to test URM walls, considering different vertical pre-compression stresses (i.e., 0.7 MPa, 1.4 MPa and 2.8 MPa). These chosen values represented 5%, 10% and 20% of the masonry compression strength as determined in the prism tests. Subsequently, a cyclic shear load was administered to each specimen, with a load speed of 1 mm/min for small displacement increments and 5 mm/min for larger displacement increments. The prescribed cyclic shear load was imposed via computer-controlled displacement steps.

It is worth noting that the testing results showed significant variations in the cyclic behaviour of the specimens based on the level of pre-compression stress, the presence of the DPC and the inherent uncertainty of masonry properties. At the lowest pre-compression level, the wallette without DPC (Series C) exhibited rocking behaviour. At the two higher pre-compression levels, the Series C wallettes experienced compression failure (toe-crushing) with minimal energy dissipation, resulting in the development of vertical cracking. The crack patterns for each Series C test are presented in Figure 4.

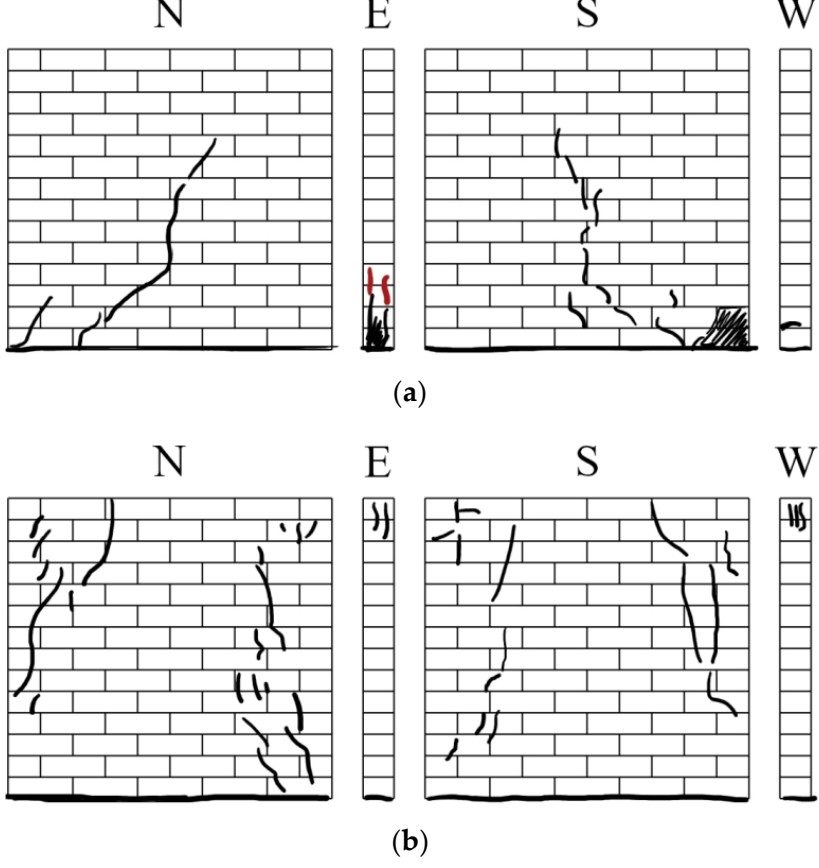

**Figure 4.** *Cont.*

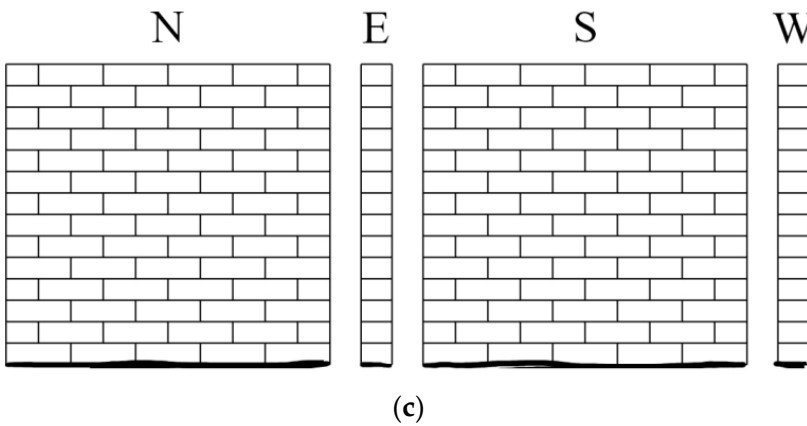

(c)

**Figure 4.** Representation of the crack pattern of URM walls without using DPC: (**a**) C1, (**b**) C2, (**c**) C3—Rocking.

More ductile failure modes were observed in the next set of testing of URM walls, where DPC was placed between the concrete beam and the masonry (Series B). At the lowest pre-compression level, the Series B wallettes demonstrated a sliding mechanism. When the pre-compression was increased to 1.4 MPa, some compression failure occurred. While at the highest pre-compression level, the Series B wallettes failed through toe-crushing. The findings of this experimental campaign showed that introducing a DPC layer increased energy dissipation, leading to quasi-ductile behaviour. The depicted crack patterns are illustrated in Figure 5. In the following section, the discussed experimental campaign is simulated using the proposed discontinuum-based modelling approach, and the results obtained are discussed thoroughly.

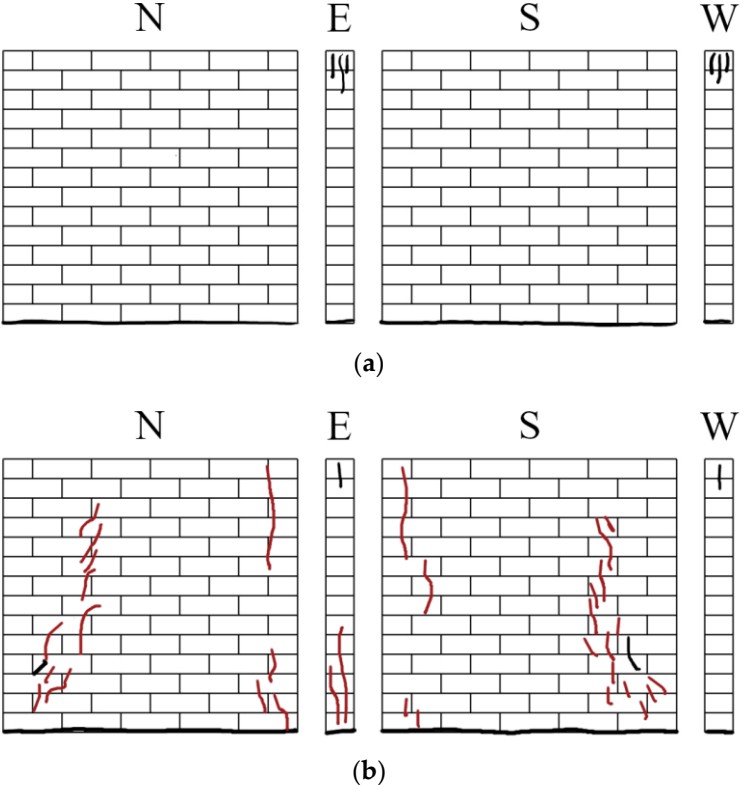

**Figure 5.** *Cont*.

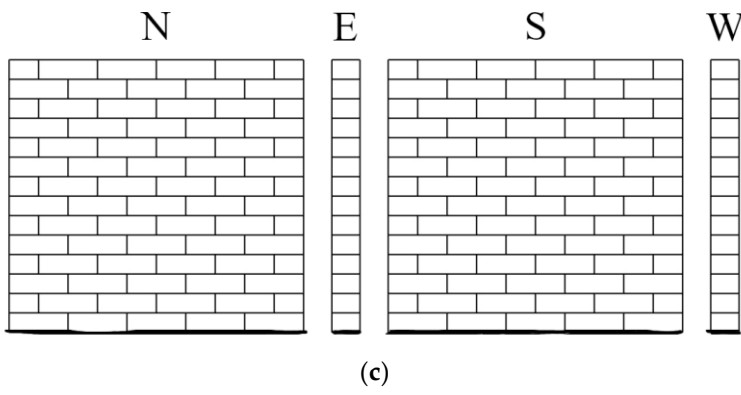

(**c**)

**Figure 5.** Representation of the crack pattern of URM walls with DPC: (**a**) B1, (**b**) B2, (**c**) B3—Sliding failure.

## 4. Results: Computational Models vs. Experiments

The adopted DEM-based modelling strategy utilizes rigid blocks where all possible failure mechanisms (i.e., cracking, sliding and crushing) are simulated at the contact points, as mentioned earlier. To replicate the benchmark study, a total of 153 discrete blocks with nearly 6000 contact points are utilized in the computational model. Once the boundary conditions are assigned, the numerical model is subjected to both gravitational acceleration and vertical pre-compression pressure. Note that the support block at the bottom is fixed, and the loading beam is set free in translation and rotation. Then, lateral displacement history is imposed on the top beam based on the given loading protocol in the reference study, which consists of horizontal displacement in the form of a sinusoidal wave, and each step is repeated three times (see [34]). The proposed discontinuum model is shown in Figure 6. Within the adopted modelling framework, different contact properties are defined among the rigid blocks to capture masonry unit (brick) and unit–mortar interface interactions. As illustrated in Figure 6, the brick failure is only represented at the potential crack surface prescribed at the mid-length of each brick, whereas the bond fracture (in tension and shear) and crushing of the masonry composite are simulated at the contact points defined along the contact surfaces between masonry units. Table 2 presents elastic and plastic contact properties used for bonds and bricks. The normal and shear contact stiffnesses defined for masonry units ($k_{n,u}$ and $k_{s,u}$) are determined as five times larger than the corresponding bond stiffnesses ($k_{n,j}$ and $k_{s,j}$), which ensures that the soft mortar–stiff masonry unit combination discussed in the referenced study is obtained [35]. The adopted strength parameters for bricks ($f_{t,u}$, $c_u$, $\phi_{0,u}$, $\phi_{res,u}$) and unit–mortar interfaces ($f_{t,j}$, $c_j$, $f_m$), together with the mode-I and compression fracture energies, are taken from the available material characterization tests or in line with recommended values in the literature [29,36,37]. Furthermore, mode-II fracture energy is predicted using an empirical relationship (see Table 2) where the dissipated energy in shear is calculated as a function of normal stress, presented in [38]. It is important to recall that the compression failure of masonry composites is only addressed at the zero-thickness interfaces, allowing inter-block penetration.

**Table 2.** Contact properties used for bond (unit–mortar interface) and masonry units (compression stress $\sigma < 0$).

| Contact Stiffness | Plastic Contact Properties | | |
|---|---|---|---|
| $k_{n,j}, k_{s,j}$ (GPa/m) | $f_{t,j}, c_j, f_m$ (MPa) | $\phi_{0,j}, \phi_{res,j}$ (°) | $G_f^I, G_f^{II}, G_c$ (N/m) |
| 15, $0.4k_{n,j}$ | 1.0, 1.4, 14.2 | 30, 30 | 15, $0.058–0.13\sigma$, $1.2f_m$ |
| $k_{n,u}, k_{s,u}$ (GPa/m) | $f_{t,u}, c_u$ (MPa) | $\phi_{0,u}, \phi_{res,u}$ (°) | $G_f^I, G_f^{II}$ (N/m) |
| 75, $0.4k_{n,u}$ | 2.8, $1.4f_{t,u}$ | 55, 55 | 42, 630 |

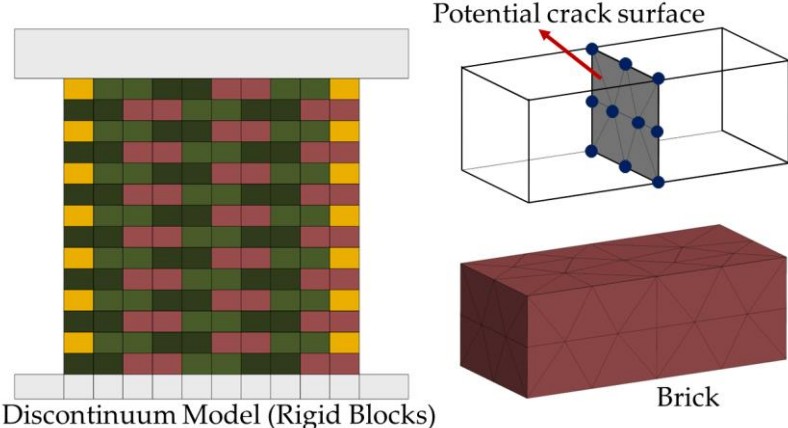

**Figure 6.** Illustration of the proposed computational model, including a brick and the potential crack surface defined within and between the masonry units (bricks).

First, the benchmark URM walls with no DPC membrane are simulated following the proposed discontinuum-based modelling approach. Two loading conditions, monotonic and cyclic (following the identical loading protocol performed during the experiment), are considered for each wall, denoted as C1, C2 and C3. It is important to recall that each URM wall has different vertical pre-compression loads (shown previously in Table 1), which play an important role in the overall behaviour and capacity of the URM panels. As shown in Figure 7, a good agreement is found between the numerical predictions and test results considering the maximum load-carrying capacity and overall deformability of the system.

In the adopted modelling strategy, the macro stiffness degradation is governed by the pre-determined local mechanisms taking place at the contact points among the discrete rigid blocks. Therefore, the computed stiffness degradation observed in the loading and unloading force cycles is driven by the local stiffness degradation and/or the loss of contact points in the numerical model. Overall, the predicted cyclic response matches reasonably well with the experimental findings regarding the URM walls with no DPC membranes subjected to different vertical pre-compression loads. However, the results of the presented pushover analysis slightly overestimate the ultimate load for all three tests, as shown in Figure 7a–c. Although this observation is expected due to the limitations of imposed monotonic loading, it underlines the effect of the adopted structural analysis methodology and the inherent limitation of the pushover analysis. The difference between the predicted load-carrying capacity obtained from pushover analysis and cyclic loading gets more noticeable for higher vertical pressures (see the difference between Figure 7b,c), and the behaviour of the URM walls becomes less ductile, which can be depicted from both monotonic and cyclic loading conditions. Note that in pushover analysis, the imposed lateral displacement at the loading beam is interrupted once the base shear force is dropped below 80% of the ultimate load [39].

The damage evolution noted during monotonic and cyclic loading analysis for URM panels with no DPC membrane is shown in Figures 8 and 9, respectively. The damage state of the contact points is investigated closely by separating the scaler damage parameter for tension ($d_t$), shear ($d_s$) and compression ($d_c$), denoted in Figures 8 and 9. The results of the pushover analyses indicate very minor toe-crushing for low pre-compression and moderate compression failure for the other two pre-compression loads accompanied by the rocking mechanism of the URM panel, which aligns with the experimental findings. It is also noted that significant damage in tension occurs at the bed joints at the side of flexural tensile stresses of the wall cross-section, while the sliding mechanism is captured in a localized area at the bottom course of the masonry walls (see Figure 8). Moreover, it is interesting to notice that shear failure at the contact points shifts toward the tension section of the wall cross-section for higher vertical pre-compression loads where the compression damage occurs at more contact points around the toe. In general, similar behaviour is observed in

pushover analysis from the proposed numerical investigations, predominately governed by toe-crushing and the flexural/rocking behaviour of the URM wall panels.

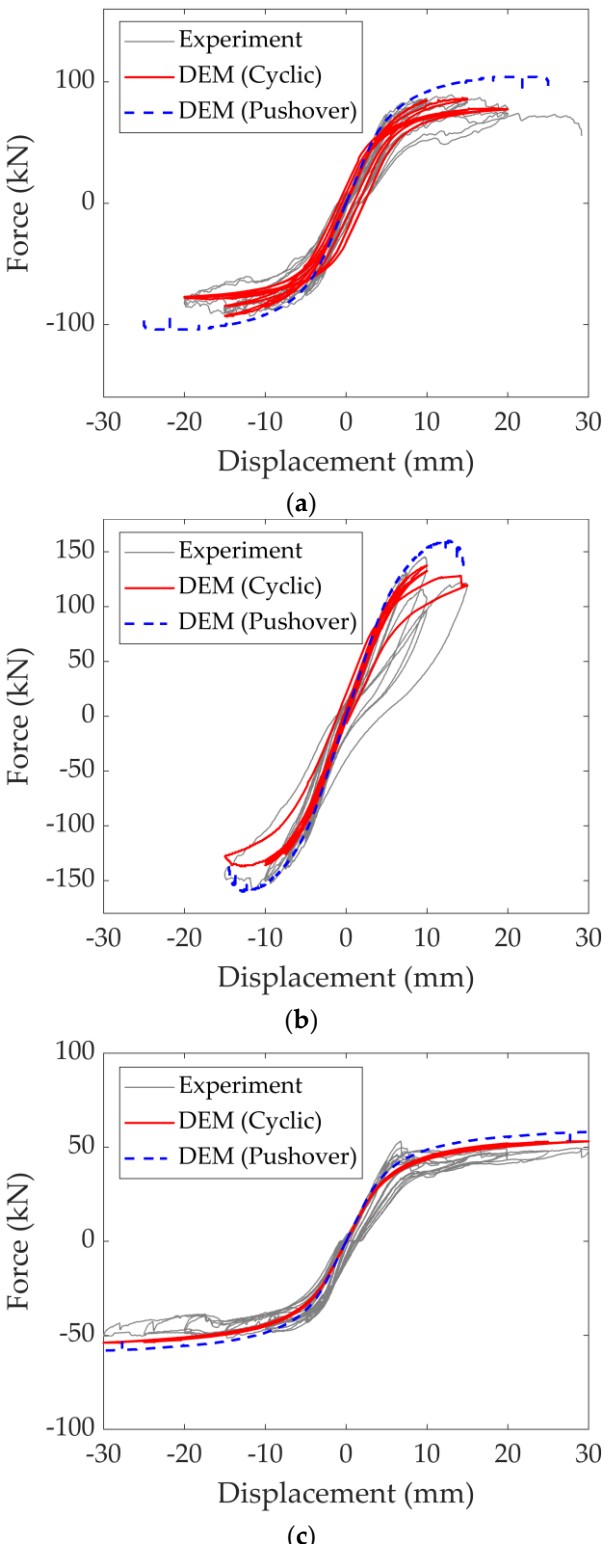

**Figure 7.** Comparing the results of the computational model (providing pushover and cyclic loading scenarios) against the experimental findings—URM walls without DPC membranes: (**a**) C1, (**b**) C2, (**c**) C3.

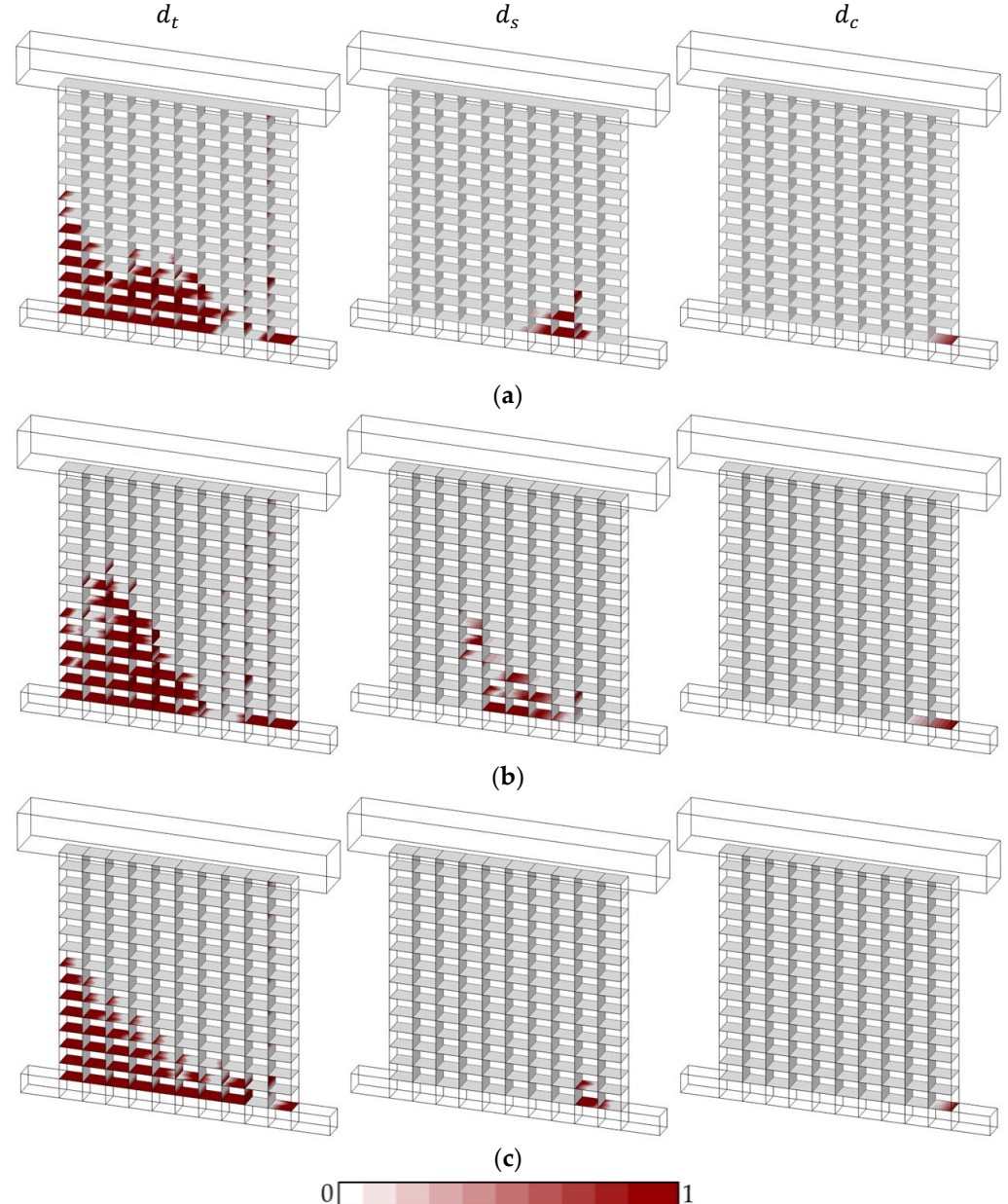

**Figure 8.** Predicted damage pattern from discontinuum-based analysis of URM walls without DPC membranes under monotonic lateral loading (pushover): (**a**) C1, (**b**) C2 and (**c**) C3—No damage: 0, Full damage: 1.

Unlike monotonic loading, rather diverse failure mechanisms and damage patterns are obtained from cyclic loading, as shown in Figure 9. For low pre-compression pressure (C3—0.7 MPa), a relatively similar response compared to pushover analysis is obtained, where cracking at the bed joints is noted without any diagonal shear crack and toe-crushing (Figure 9c). For moderate vertical pre-compression (C1—1.4 MPa), minor crushing together with a significant sliding mechanism is predicted (Figure 9a). Finally, a vertical crack passing through the bricks and head joints is obtained for high pre-compression pressures (C2—2.8 MPa) with toe-crushing, as shown in Figure 9b. The predicted cracking mechanism can be compared with the experimental results (Figure 4b); however, the proposed modelling approach could not capture multiple nearly vertical crack lines, as noted during the testing.

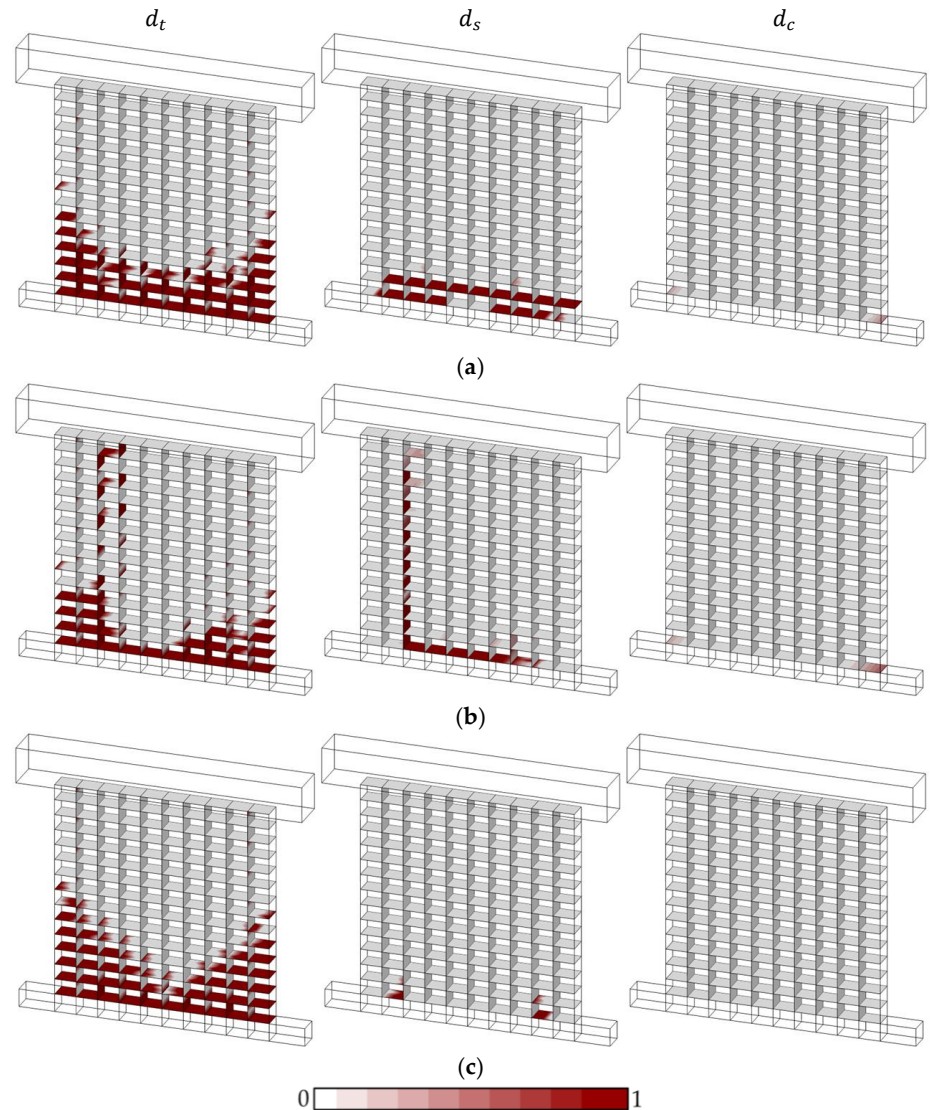

**Figure 9.** Predicted damage pattern from discontinuum-based analysis of URM walls with DPC membrane under cyclic lateral loading: (**a**) C1, (**b**) C2 and (**c**) C3—No damage: 0, Full damage: 1.

Next, the URM walls with DPC membranes are analyzed. To represent the DPC membrane, the mechanical properties between the bottom course layer of masonry and the support block are modified. A lower bond strength and sliding resistance at the contact points are considered by utilizing a lower friction angle ($\phi_{0,DPC} = \phi_{res,DPC} = 18°$) in line with the available studies [40]. The results of the computational investigations (including monotonic and cyclic loading) are presented in Figure 10 and compared against the experimental results. Other than the high pre-compression load (B2—Figure 10b), the load-carrying capacity of URM panels is underestimated by the adopted discontinuum-based models, as shown in Figure 10a,c. Although a similar trend is predicted through DEM-based simulations in unloading–reloading cycles for B1 and B3, the ultimate lateral load is controlled by the sliding resistance of the mortar joints at the bottom course of the URM wall. In line with the predicted hysteric response, the results of the pushover analyses (monotonic loading) provide an identical failure mechanism, where the maximum lateral load is directly controlled by the sliding capacity, yielding an elastic–perfectly plastic response which are also depicted in the presented force–displacement curves (see Figure 10). The associated failure mechanisms of the analyzed URM walls are given in Appendix A, where the readers can find the computed damage states at the contact points for all three pre-compression loads. As mentioned earlier, the dominant mechanism,

predicted as a sliding failure, can be noticed at the bed joints for pushover and cyclic loading, illustrated in Figures A1 and A2, respectively. Since no significant difference is noted in the obtained damage patterns (from both monotonic and cyclic loading), the results are provided in Appendix A. While most of the presented observations, based on the outcomes of the computational models, are in good agreement with the experimental observations, crushing failure is noted during the experiment with high pre-compression pressure (B2) instead of sliding failure. This discrepancy may come from the obtained premature sliding failure mechanism preventing the rocking and eventual crushing failure of masonry in the numerical model. Moreover, the adopted modelling approach does not address the material uncertainty, providing a limited perspective to capture different failure modes under identical loading and boundary conditions.

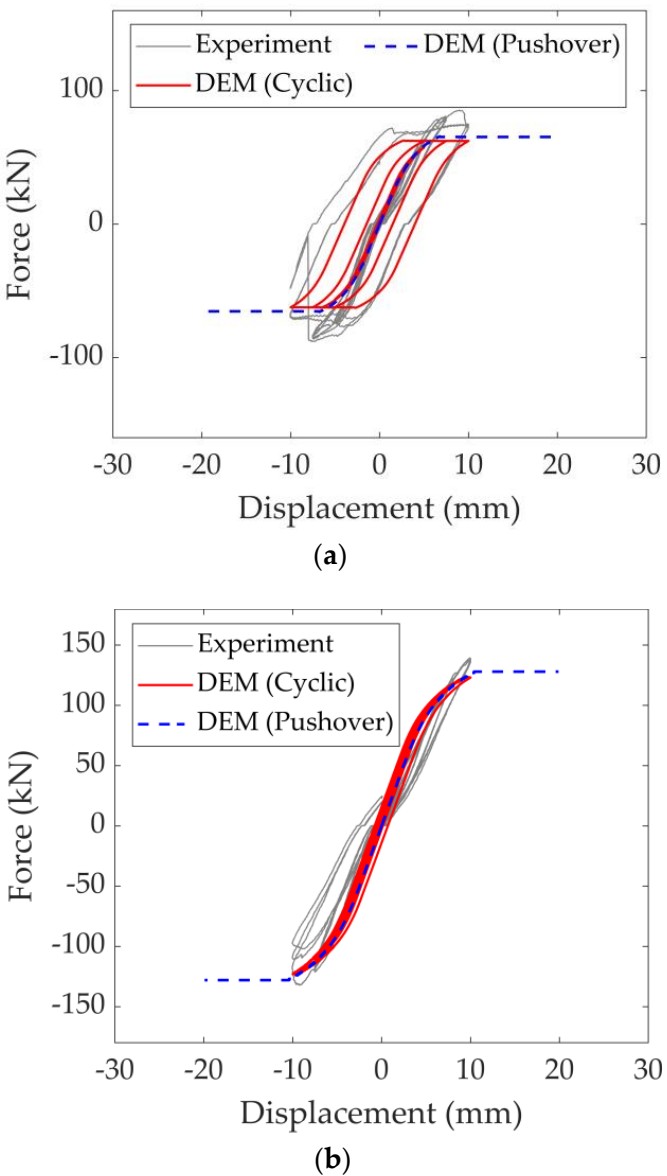

**Figure 10.** *Cont.*

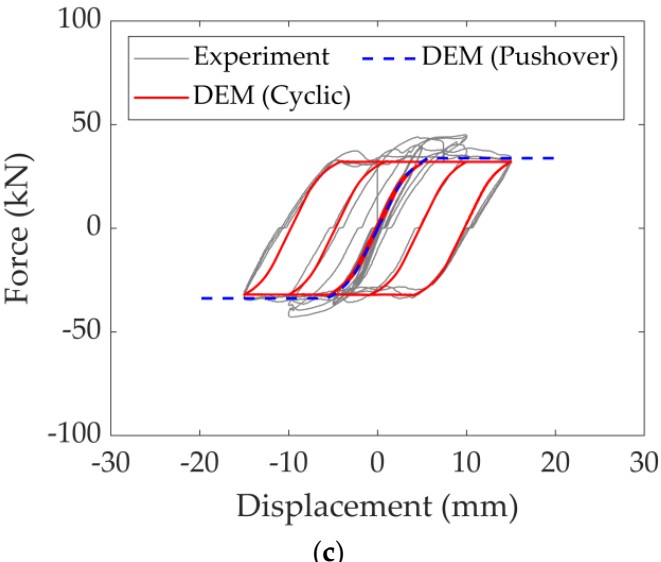

(**c**)

**Figure 10.** Comparing the results of the computational model (providing pushover and cyclic loading scenarios) against the experimental findings—URM walls with DPC membranes: (**a**) B1, (**b**) B2, (**c**) B3.

## 5. Conclusions

The present research explores the in-plane cyclic behaviour of URM walls under different pre-compression pressures through the application of a discontinuum-based modelling strategy. A bi-linear elasto-softening contact constitutive law for tension, compression and shear is utilized to address the bond behaviour within the DEM framework. Unlike commonly used standard (brittle) contact constitutive models, the adopted contact model considers stiffness degradation at the contact points in tension–compression regimes. To validate the adopted modelling approach, relying on rigid block interaction, the experimental behaviour of URM wallettes under cyclic loading with and without DPC membranes is simulated. Both monotonic lateral loading (pushover) and cyclic analyses are utilized to analyze URM walls with varying levels of pre-compression pressures. From the presented analyses, the following conclusions can be drawn.

- For walls without a DPC membrane (Series C), the adopted computational modelling strategy accurately captures the load-carrying capacity of the walls under cyclic loads. Conversely, as expected, the pushover analyses are shown to overestimate the ultimate load slightly/moderately for the three walls depending on the level of pre-compression loads. The discrepancy between the monotonic and cyclic loading conditions is found to increase at high vertical pressures. Moreover, the predicted cracking patterns in the pushover and cyclic analyses show reasonable agreement with the experimental results.
- For walls with a DPC membrane placed between the masonry wall and the concrete base (Series B), the adopted computational modelling strategy provides a good match with the experimental observations, except in specimen B2 (high precompression) denoting sliding instead of crushing failure according to the experimental findings. Furthermore, it is noted that the proposed computational models underestimate the load-carrying capacity when using a DPC membrane (25–30%) except for B2 (high precompression).
- Overall, the potential of the proposed discontinuum-based analysis is demonstrated by utilizing a novel contact model for the first time to investigate the cyclic behaviour of URM walls. The present research also underlines the necessity of considering advanced contact constitutive laws to accurately capture the local mechanisms of masonry assemblages.

- In a future study, the adopted modelling approach will be extended toward a stochastic analysis to offer a broader perspective on URM wall behaviour and capture different failure modes under identical loading and boundary conditions. Moreover, the proposed contact model will be utilized to explore the progressive failure mechanisms in URM walls subjected to cyclic loading.

**Author Contributions:** Conceptualization, B.P., J.V.L. and N.M.; methodology, B.P.; software, B.P., R.W. and J.V.L.; validation, B.P., R.W. and J.V.L.; formal analysis, R.W. and B.P.; resources, B.P.; writing—original draft preparation, B.P. and R.W.; writing—review and editing, J.V.L. and N.M.; visualization, B.P. and N.M.; supervision, N.M. and J.V.L.; project administration, B.P.; funding acquisition, B.P. All authors have read and agreed to the published version of the manuscript.

**Funding:** This research is supported by the Itasca Education Partnership (IEP)—Teaching Program.

**Data Availability Statement:** The data that support the findings of this study are available from the corresponding author upon reasonable request.

**Conflicts of Interest:** The authors declare no conflicts of interest.

## Appendix A

In the appendix, the obtained damage patterns in URM walls with DPC membranes under different pre-compression pressures are presented. Similar kinematic mechanisms are observed comparing the monotonic lateral loading (pushover) and cyclic analysis where the sliding failure governs the macro-behaviour of the walls that takes place at the interface between the membrane and the bottom course of the URM wall, shown in Figures A1 and A2, respectively.

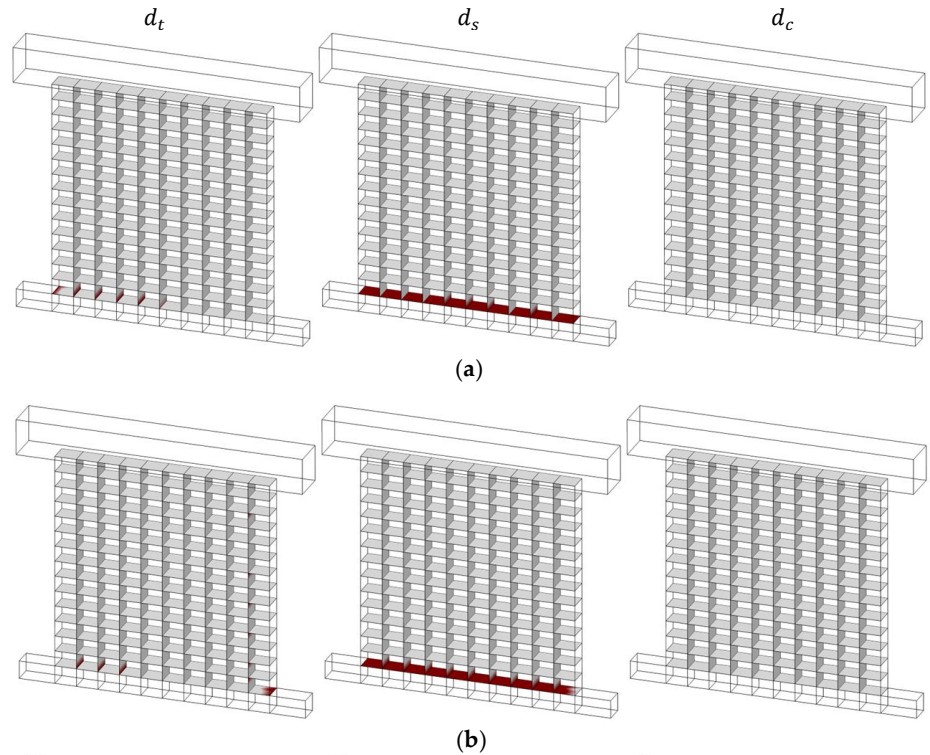

**Figure A1.** *Cont.*

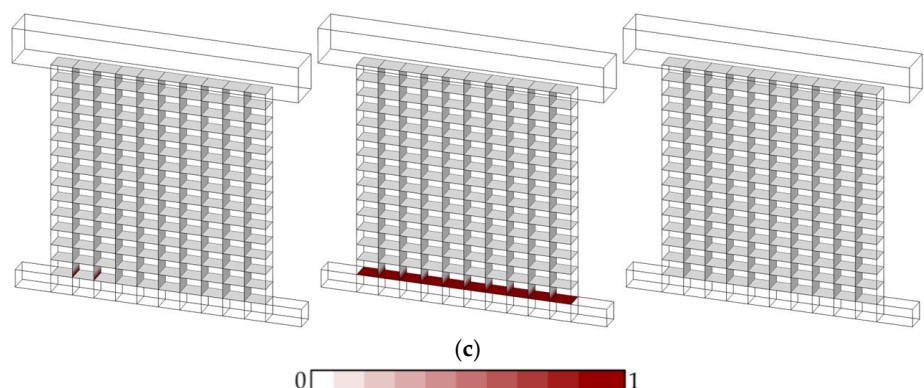

(**c**)

**Figure A1.** Predicted damage pattern from discontinuum-based analysis of URM walls with DPC membrane under monotonic lateral loading (pushover): (**a**) B1, (**b**) B2 and (**c**) B3—No damage: 0, Full damage: 1.

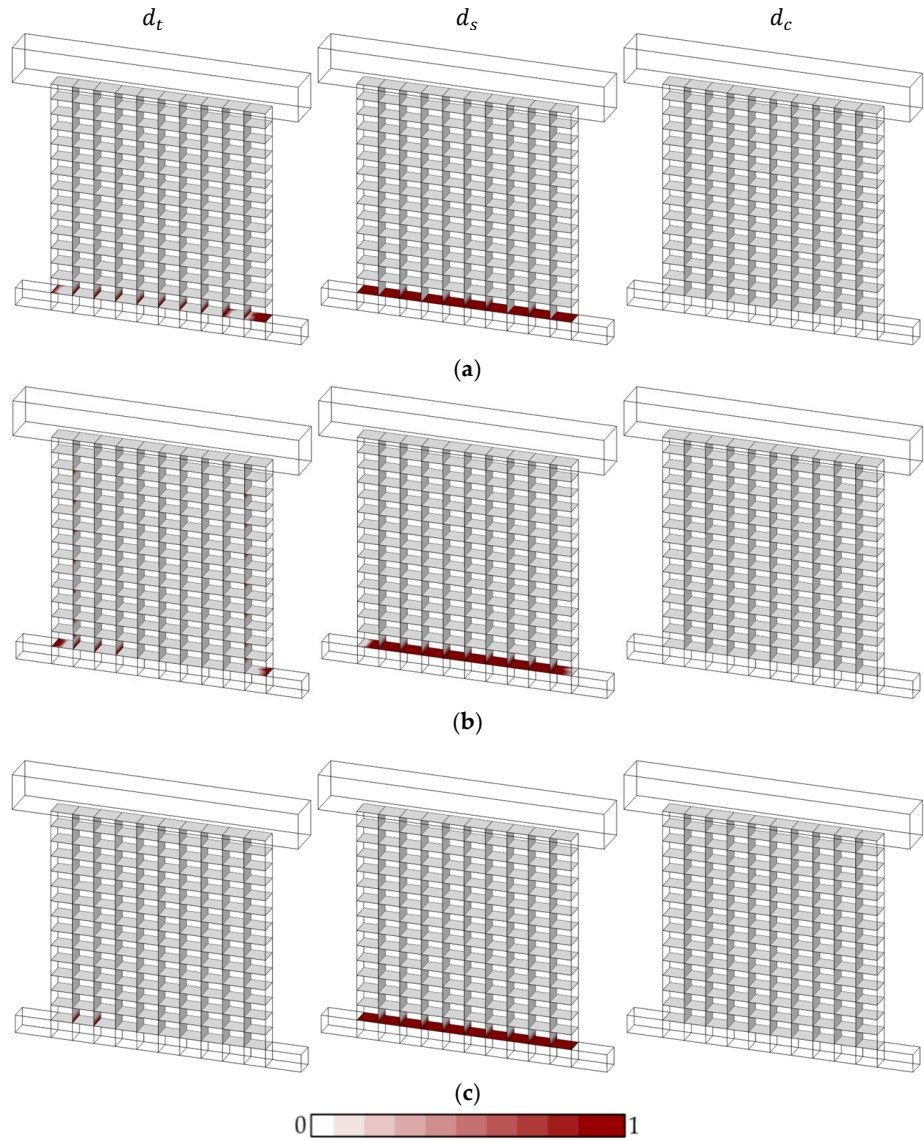

(**c**)

**Figure A2.** Predicted damage pattern from discontinuum-based analysis of URM walls with DPC membranes under cyclic lateral loading: (**a**) B1, (**b**) B2 and (**c**) B3—No damage: 0, Full damage: 1.

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
