# Peer review of "Exploring the Cyclic Behaviour of URM Walls with and without Damp-Proof Course (DPC) Membranes through Discrete Element Method"

_infrastructures, doi:10.3390/infrastructures9010011_

Round 1

Reviewer 1 Report

Comments and Suggestions for Authors

This work is good enough for the journal. I recommend it for publication.

Author Response

Thanks for the comment. The authors appreciate the time allocated for the submitted manuscript. 

Reviewer 2 Report

Comments and Suggestions for Authors

The paper “Exploring the cyclic behaviour of URM walls with and without damp-proof course (DPC) membrane through DEM” reports an interesting research work about the use of a discontinuum-based modelling strategy based on the discrete element method (DEM) to investigate the in-plane cyclic response of URM panels under different vertical pressures with and without a damp-proof course (DPC) membrane. In general, the paper appears well-written and well-organized in its different Sections. Furthermore, the approach proposed by the authors is clearly described in the text and the results obtained are of interest for the scientific community and designers. For these reasons, it is opinion of this reviewer that the manuscript can be considered for the publication in Infrastructures journal after the following minor corrections/improvements:

- Introduction in the sentence “Past and recent post-earthquake investigations have revealed the diverse failure mechanisms of URM buildings, where cracks are predominantly localized along the unit mortar interfaces due to weak tensile strength and shear resistance” consider as reported in 10.1080/15583058.2021.1904056.

- line 214: replace “Mpa” with “MPa”

- line 219: replace “Is” with “is”

- Figure 5: why are some cracks highlighted in red? Clarify this aspect.

- In the conclusions better highlight the original aspects and the novelty of the research work

- Some consideration of the further developments of the research work can be added

Author Response

Responses are attached.

Reviewer 3 Report

Comments and Suggestions for Authors

The paper introduces a modeling approach based on the discrete element method (DEM) to investigate the in-plane cyclic response of unreinforced masonry (URM) walls with and without damp-proof course (DPC) membranes under varying vertical pressures. The URM walls are conceptualized as discrete rigid block systems interacting through contact points along their boundaries. Within this discontinuum-based modeling framework, a novel contact constitutive model is employed to address the elasto-softening stress-displacement behavior of unit-mortar interfaces, accounting for stiffness degradation in tension-compression regimes. The insights derived from this study contribute to understanding the cyclic behavior of URM walls, offering potential applications in the design and analysis of structures featuring such walls. Regarding my recommendations, I propose accepting the paper with minor edits. Here are my specific comments:

1.       Introduction: The introduction is comprehensive, referencing relevant literature, and effectively emphasizing research limitations, gaps, and novelty.

2.       Line 62: I would suggest replacing the term "smear out" with a more technical expression.

3.       Lines 135-148: I appreciate the emphasis on highlighting the limitations of the existing DEM-based computational model.

4.       Lines 216-218: I need clarification on the applied load and whether the test employed load or displacement control. Detailed information on the test loading conditions is necessary.

5.       Figure 4. c and Figure 5.c: Can you clarify whether these figures depict a sliding mechanism? I'm also interested in information on the associated crack patterns.

6.       Lines 250-252: Could you specify the imposed displacement in the context of the discussed content?

7.       Lines 292-296: I would appreciate further elaboration on the mentioned points for enhanced clarity.

8.       Lines 300-302: I'm curious about the rationale behind the assumption made in this context.

9.       Comparison to Other DEM Methods: It might be beneficial to include a comparison to other DEM methods to assess model accuracy.

10.   Figures 7.a, 7.b, 10.a, 10.b: I've noticed that the numerical model does not accurately capture the tilting of the hysteresis response over time. Could you elaborate on this phenomenon?

Comments on the Quality of English Language

The English quality is good. Minor edits might be required.

Author Response

Responses are attached.
